# Long-Term *SMN*- and *Ncald*-ASO Combinatorial Therapy in SMA Mice and *NCALD*-ASO Treatment in hiPSC-Derived Motor Neurons Show Protective Effects

**DOI:** 10.3390/ijms24044198

**Published:** 2023-02-20

**Authors:** Anixa Muiños-Bühl, Roman Rombo, Karen K. Ling, Eleonora Zilio, Frank Rigo, C. Frank Bennett, Brunhilde Wirth

**Affiliations:** 1Institute of Human Genetics, University of Cologne, 50931 Cologne, Germany; 2Center for Molecular Medicine Cologne, University of Cologne, 50931 Cologne, Germany; 3IONIS Pharmaceuticals, Carlsbad, CA 92010, USA; 4Center for Rare Diseases Cologne, University Hospital of Cologne, 50931 Cologne, Germany

**Keywords:** antisense oligonucleotide, genetic modifier, hiPSCs, neuromuscular disorder, NCALD, SMA, *SMN2*, therapy

## Abstract

For SMA patients with only two *SMN2* copies, available therapies might be insufficient to counteract lifelong motor neuron (MN) dysfunction. Therefore, additional SMN-independent compounds, supporting SMN-dependent therapies, might be beneficial. Neurocalcin delta (NCALD) reduction, an SMA protective genetic modifier, ameliorates SMA across species. In a low-dose *SMN*-ASO-treated severe SMA mouse model, presymptomatic intracerebroventricular (i.c.v.) injection of *Ncald*-ASO at postnatal day 2 (PND2) significantly ameliorates histological and electrophysiological SMA hallmarks at PND21. However, contrary to *SMN*-ASOs, *Ncald*-ASOs show a shorter duration of action limiting a long-term benefit. Here, we investigated the longer-term effect of *Ncald*-ASOs by additional i.c.v. bolus injection at PND28. Two weeks after injection of 500 µg *Ncald*-ASO in wild-type mice, NCALD was significantly reduced in the brain and spinal cord and well tolerated. Next, we performed a double-blinded preclinical study combining low-dose *SMN*-ASO (PND1) with 2× i.c.v. *Ncald*-ASO or CTRL-ASO (100 µg at PND2, 500 µg at PND28). *Ncald*-ASO re-injection significantly ameliorated electrophysiological defects and NMJ denervation at 2 months. Moreover, we developed and identified a non-toxic and highly efficient human *NCALD*-ASO that significantly reduced NCALD in hiPSC-derived MNs. This improved both neuronal activity and growth cone maturation of SMA MNs, emphasizing the additional protective effect of *NCALD*-ASO treatment.

## 1. Introduction

Spinal muscular atrophy (SMA), a common and devastating neuromuscular disorder, has changed from an early lethal to an efficiently treatable disease for a high proportion of patients in the last years [1,2,3,4,5]. SMA, an autosomal-recessive inherited neurodegenerative disease, has an incidence of 1 in 6000–10,000 with a carrier frequency of 1:51 worldwide and 1:41 in the European population [6]. SMA is caused by biallelic mutations in the survival motor neuron 1 (*SMN1)* gene, with all patients retaining *SMN2,* an almost identical copy gene [6]. Approximately 96% of SMA patients show homozygous deletions of exon 7 and 8 or only of exon 7 of *SMN1*, making genetic testing fast and neonatal screening highly reliable (reviewed by [7,8]). Due to a translationally silent mutation in exon 7 of the *SMN2* gene that affects correct splicing, only small amounts of full-length mRNA are produced, leading to a marked decrease in functional SMN protein [6,9,10]. 

In SMA, the disease severity inversely correlates with the copy number of *SMN2*; the more, the better the phenotype. Approximately 50% of SMA patients develop SMA type 1; the majority of them carry only two *SMN2* copies, are unable to sit or walk and die before 2 years of age. Around 30% of patients have SMA type 2, an intermediate phenotype with patients achieving the ability to sit independently, who usually carry three *SMN2* copies. Another ~20% of SMA patients have SMA type 3 with a mild SMA phenotype. These patients are able to walk independently; however, they often become wheelchair-bound with the progression of the disease. SMA type 3 patients normally carry 3–4 *SMN2* copies. Rarely, patients, who carry 4–6 *SMN2* copies can develop an adult SMA type 4 [11,12]. 

SMN is a ubiquitously expressed protein with variable expression patterns in different tissues, but especially abundant in the lower spinal cord motor neurons (MNs) [13,14]. SMN is a multifunctional protein involved in key cellular processes such as snRNP assembly and splicing, mRNA transport, microRNA biogenesis, local translation, cytoskeleton dynamics, endocytosis, DNA damage repair; mitochondrial biogenesis and others [15,16,17]. Although SMA is considered a MN disorder, the housekeeping functions of SMN explains the multiorgan disfunction in its severe forms, when SMN levels are reduced below a certain threshold [18]. However, why MNs are predominantly vulnerable to reduced SMN protein remains largely unclear. SMN expression levels vary between tissues and stages of development [19], but are particularly required during neuromuscular junction (NMJ) maturation [20]. Moreover, SMN binding to ribosomes and polysomes occurs in a tissue-specific manner in vivo [21] and its deficiency leads to tissue-specific splicing defects that might contribute particularly to MN vulnerability [22]. These findings point towards differential requirements of SMN in each tissue and developmental stage.

Three highly effective SMN-dependent therapies have been developed using gene therapy, antisense oligonucleotides or small molecules (reviewed in [7,8]). If the therapy is applied presymptomatically—requiring neonatal screening—children with three and more *SMN2* copies meet age-appropriate developmental milestones. Instead, children with only two *SMN2* copies show delayed or deficient motor milestones [23,24,25,26]. Therefore, additional SMN-independent therapies are more than ever necessary to complement SMN-dependent therapies. Long-term treatment of symptomatic individuals with SMA is extremely costly, but so are current therapies, therefore every effort must be made not only to treat but to cure SMA [7,27].

The identification of SMA discordant families with asymptomatic individuals carrying homozygous *SMN1* deletions and three to four *SMN2* copies led to the discovery of two SMA genetic modifiers, plastin 3 (PLS3) and neurocalcin δ (NCALD), in humans [28,29]. These findings not only opened the door to a better understanding of crucial molecular pathways disturbed in SMA, but also to the development of combinatorial therapies [29,30,31]. PLS3 is a F-actin binding and bundling protein, and NCALD a neuronal calcium sensor; both act on multiple cellular pathways in MNs including axonal growth, calcium homeostasis, neurotransmission, endocytosis and others [28,29,32,33]. PLS3 overexpression or NCALD reduction ameliorate SMA hallmarks at MN and NMJ level and prolong survival. The protective effect of both modifiers was corroborated in various genetically modified SMA animal models and by use of gene therapy or antisense oligonucleotides [30,31,32,34,35,36]. A first randomized-blinded preclinical study in SMA mice testing an antisense oligonucleotide-based combinatorial therapy targeting *SMN* and *Ncald*, showed a synergistic amelioration of SMA hallmarks, such as electrophysiological and morphological properties of NMJs and muscles at postnatal day (PND) 21 [30]. Interestingly, the effect of *Ncald*-ASO was rather short and thus comparable to what has been described in a similar therapeutic approach using *Chp1*-ASO [37], a PLS3 interacting partner and SMA modifier [38], which might explain the failure of a long-term amelioration [30].

In light of these encouraging results, the aim of this study was to determine if *Ncald*-ASO re-injections could prolong the therapeutic effect observed at PND21 in SMA mice. In addition, we developed human *NCALD*-ASOs to be tested for efficacy and non-toxicity, and investigated their impact on MN development and neuronal function. Human MNs were differentiated from control and SMA patient-derived inducible pluripotent stem cells (hiPSCs). For both conditions, SMA mice and human MNs, we observed a positive effect of NCALD reduction on MN function, further supporting the protective role of NCALD reduction for combinatorial SMA therapies.

## 2. Results

### 2.1. Re-Injection of Ncald-ASO Significantly Reduces NCALD in Brain and Spinal Cord

In order to mimic the situation of *SMN1*-deleted individuals, who remain asymptomatic despite carrying only four *SMN2* copies, we injected at PND1 a suboptimal low dose of 30 µg of *SMN*-ASOs into the severely-affected Taiwanese SMA mice that carry only two human *SMN2* copies on null murine background, to produce a mildly affected SMA model that resembles the human situation (Figure 1A) [29,30,37]. As previously demonstrated, this suboptimal *SMN*-ASO treatment significantly prolongs survival by ameliorating the multiorgan impairment, without fully restoring MN and NMJ function [29,30,31,37], allowing the study of protective modifiers in combinatorial therapies. Since a single *Ncald*-ASOs injection at PND2 had a positive effect on SMA hallmarks but turned out to be rather unstable after 3–4 weeks [30], here, we investigated the longer-term therapeutic effect by reinjecting *Ncald*-ASOs at one month and studying the mice at two months (Figure 1B). 

First, we tested the efficacy and tolerability of i.c.v. bolus injection of 500 µg of CTRL-ASO and *Ncald*-ASO in adult wild-type mice. For this purpose, animals were injected at PND28 and brain and spinal cord tissues were collected 2-weeks after injection for western blot analysis. In brain lysates, NCALD protein levels were reduced to 43% compared to CTRL-ASO injected animals (Figure 1C) and in spinal cord to 63% (Figure 1D). The less pronounced reduction of NCALD in the spinal cord compared to the brain, might be due to the i.c.v. injected ASOs that have to be distributed in the CNS through the cerebrospinal fluid (CSF). Importantly, 500 µg of CTRL- and *Ncald*-ASO were well tolerated by the animals and no adverse effects were observed. 

Because *Ncald*-ASOs applied by i.c.v. bolus injection efficiently knocked down NCALD levels in spinal cord of wild-type animals, we designed a double-blind pre-clinical study in SMA and HET mice, as shown in Figure 1B. In detail, at PND1 all animals received 30 µg *SMN*-ASOs by subcutaneous injection. At PND2, half of the animals received 100 µg CTRL-ASO, the other half 100 µg *Ncald*-ASO by i.c.v. injection as previously described [30]. At PND28, animals were anesthetized and fixed in a stereotaxic equipment and received an i.c.v. bolus re-injection with 500 µg CTRL-ASO or *Ncald*-ASO (Figure 1B). One month after CTRL-ASO or *Ncald*-ASO re-injection, we determined NCALD downregulation in brain lysates of HET *Ncald*-ASO (Figure 1E,F) and SMA *Ncald*-ASO-treated animals (Figure 1E,G). Both HET and SMA mice exhibited a significant reduction of NCALD levels when compared to their respective control groups. Moreover, NCALD was significantly reduced in the spinal cord of HET animals 4-weeks after receiving *Ncald*-ASO re-injection compared to mice treated with CTRL-ASO (Figure 1H,I), however, was more variable from one animal to another in SMA mice (Figure 1H,J).

### 2.2. Long-Term Combinatorial Treatment with Ncald- and SMN-ASOs Ameliorates Electrophysiological Defects and NMJ Denervation in SMA Mice

Electrophysiological defects have been described in SMA patients and animal models, including reduction of compound muscle action potential (CMAP) and motor unit number estimation (MUNE) [30,39,40]. One single injection of *Ncald*-ASO at PND2 significantly increased both electrophysiological parameters of the gastrocnemius muscle in SMA mice at PND21, but was not sufficient to prolong the effect for a long-term [30]. Here, SMA mice re-injected with *Ncald*-ASO showed a significant amelioration of the CMAP response, indicating that prolonged NCALD reduction has a beneficial effect on the functionality of the motor units (Figure 2A,B). Moreover, MUNE analysis confirmed reduced motor unit numbers in SMA compared to HET CTRL-ASO-treated animals, a parameter that was rescued upon *Ncald*-ASO re-injection (Figure 2C). This data strongly supports that 500 µg *Ncald*-ASO re-injection in combination with a single low-dose *SMN*-ASO administration not only prolongs the motor unit function but also prevents the loss of motor unit numbers in SMA mice over time, thus protecting from neurodegeneration. 

Since previous work showed that one single injection of *Ncald*-ASO administered at PND2 improved pathological features of the NMJ in 3-week-old mice but failed to show any amelioration at 3 months [30], we analyzed, here, the NMJ area and innervation of the transversus abdominis muscle upon *Ncald*-ASO re-injection. To quantify the NMJ area, the size of bungarotoxin (BTX)-positive endplates was measured. To analyze NMJ innervation, NMJs were categorized according to the percentage of nerve terminal that overlaid with the acetylcholine receptors (AChRs) as fully innervated (>80%), partially innervated (>15–79%) and non-innervated (<15%). NMJ area and innervation were affected in SMA mice compared to HET littermates. Strikingly, re-injection with 500 µg *Ncald*-ASO did not increase the NMJ area in SMA animals compared to CTRL-ASO injected SMA mice at 2 months of age (Figure 2D). However, denervation of NMJs was ameliorated upon *Ncald*-ASO re-injection in SMA animals (Figure 2E), emphasizing the key role of NCALD reduction in preventing the progressive retraction of the nerve terminal from its target muscle fiber, hence halting neurodegeneration.

Finally, we analyzed muscle fiber size from gastrocnemius muscles at 2 months of age. Muscle fiber size grouping did not reveal an increase in fiber size upon 500 µg *Ncald*-ASO reinjection in SMA animals compared to CTRL-ASO SMA mice (Figure 2F).

### 2.3. Testing of Human NCALD-ASOs in MNs Derived from SMA and Control hiPSCs 

Based on our positive results with *SMN*- and *Ncald*-ASO long-term combinatorial therapy in SMA mice, we next aimed to develop a similar approach for humans. Therefore, a full battery of human *NCALD*-ASO were designed and synthesized, which were tested for non-toxicity and efficiency in various cellular systems. Next, the three top hits were investigated for toxicity and tolerability in MNs derived from SMA and control hiPSCs (Table 1 and Appendix A). The MN differentiation protocol was based on dual SMAD inhibition (Appendix A) carried out in two SMA type I lines (HGK1 and CS32iSMA) and two control hiPSC (HUVEC and WTC11) lines. At day 15 of the differentiation, MN cultures were composed of about 60% MNs, as verified by staining with the MN marker ISL1 (Appendix A). Next, we tested the efficacy and tolerability of *NCALD*-ASO candidates in hiPSC-derived MNs. *NCALD*-ASOs were tested simultaneously in control MNs (HUVEC) derived from the same differentiation, in order to detect possible side effects of the ASOs. MN cultures were treated at day 13 with 60 nM of each *NCALD*-ASO and proteins were collected at day 20 to assess NCALD reduction by Western blot. At day 20 of the differentiation, all *NCALD*-ASOs significantly downregulated NCALD levels in control MNs as determined by Western blot (Figure 3A,B). Treatment with 60 nM of *NCALD*-ASO55 and *NCALD*-ASO69 had comparable knockdown efficiencies, reducing NCALD to 51% and 46%, respectively (Figure 3B).

Treatment with 60 nM of *NCALD*-ASO89 further reduced NCALD to 30% in control MNs (Figure 3B). Surprisingly, the protein amount of the MN marker ISL1 was significantly reduced upon treatment with 60 nM of *NCALD*-ASO55 and *NCALD*-ASO89 in comparison to *CTRL*-ASO and *NCALD*-ASO69 treated MNs (Figure 3A,C). Moreover, SMN was also reduced after *NCALD*-ASO55 and *NCALD*-ASO89 treatment when compared to CTRL-ASO or *NCALD*-ASO69-treated MNs (Figure 3A,D) suggesting some off-target effects that produce toxicity in the neurons. This supports the use of 60 nM *NCALD*-ASO69 to assess the therapeutic effect of NCALD reduction in hiPSC-derived MNs. 

Next, we tested the efficiency of 60 nM *NCALD*-ASO69 in the other hiPSC-derived MNs, where no significant reduction of ISL1 protein was detected (Figure 3E–G). In addition, NCALD reduction was detected in both soma and growth cones of SMA MNs (Appendix A). Thus, we identified *NCALD*-ASO69 as an efficient and non-toxic ASO that enables NCALD reduction in all four tested hiPSC-derived MNs. Moreover, NCALD downregulation can be detected at the growth cones of the MNs, which are SMA relevant sites. To test the therapeutic potential of NCALD reduction in MNs derived from SMA hiPSCs, we decided to treat the neuronal cultures with 60 nM of *NCALD*-ASO69 at day 13, and perform the functional and morphological experiments at day 20 of the differentiation. 

### 2.4. Treatment with NCALD-ASO69 Influences Growth Cone Morphology

SMN deficiency leads to axonal outgrowth defects, smaller growth cones and defects in local translation of different actin isoforms, resulting in impaired actin dynamics that affect the presynaptic terminal [43,44]. Interestingly, NCALD is highly abundant in spinal MNs [45] and its reduction promotes neurite outgrowth in SMN-deficient neuronal cells even in the absence of retinoic acid, and axonal growth in Smn-depleted zebrafish [29]. Importantly, NCALD directly binds to actin and tubulin, two major components of the cytoskeleton [46]. Here, we investigated if NCALD reduction might have an impact on cytoskeleton dynamics at the growth cone of hiPSC-derived MNs.

First, growth cone structure was classified based on actin morphology in three categories: blunt, filopodia and lamellipodia (Figure 4A) [47]. SMA type I MNs (HGK1 and CS32iSMA) exhibited more growth cones with blunt actin filaments compared to the control lines (Figure 4A). Instead, upon *NCALD*-ASO69 treatment, more growth cones displayed filopodial and lamellipodial actin morphologies and thus were more mature. Surprisingly, *NCALD*-ASO69 also had an effect in growth cone morphology in one of the control lines (HUVEC), suggesting the NCALD might have a role in actin dynamics independently of the SMA background. Interestingly, no effect was detected in MNs derived from the control WTC11 hiPSC line, suggesting that the effect of NCALD in actin dynamics might differ between individuals.

Second, microtubule morphology was categorized based on whether the microtubules of the central domain displayed a bundled, spread or looped conformation (Figure 4B) [48]. Analysis of the microtubule conformation resulted in marked presence of bundled microtubules in the SMA type I lines compared to the control MNs treated with CTRL-ASO. The number of bundled microtubules was drastically reduced in favor of spread and looped microtubules in the control and SMA lines upon *NCALD*-ASO69 treatment. These data strongly support the role of NCALD reduction on actin and microtubule cytoskeleton modulation in SMA and therefore, growth cone dynamics, which is crucial for neurite outgrowth and NMJ maturation and function. Moreover, these results are in line with our findings on the improvement of NMJ innervation upon *Ncald*-ASO treatment in SMA mice, highlighting the potential of NCALD reduction as a promising therapeutic approach that targets one of the key pathological features in SMA, the NMJ.

### 2.5. Treatment with NCALD-ASO69 Increases Neuronal Activity

To assess the therapeutic potential of 60 nM *NCALD*-ASO69 treatment in hiPSC- derived MNs from SMA patients, we recorded the spontaneous firing of treated control and SMA type I hiPSC-derived MNs at day 20 of differentiation using a multi-electrode array system (MEA). As expected, spontaneous neuronal activity determined by the number of spikes was reduced in SMA type I MNs compared to healthy control MNs treated with CTRL-ASO (Figure 5A,B and Appendix A). Importantly, both SMA MN lines treated with *NCALD*-ASO69 exhibited an increase in number of spikes, demonstrating the protective effect of NCALD downregulation on neuronal activity. In addition, SMA MN cultures had fewer active electrodes than the control MN lines, a parameter that was improved after *NCALD*-ASO69 treatment (Appendix A). Moreover, the number of bursts were markedly reduced in SMA MNs compared to the healthy control lines (Figure 5B), a parameter that was ameliorated upon *NCALD*-ASO69 treatment.

Interestingly, an increase in active MNs was observed after *NCALD*-ASO69 treatment independently of the genotype, as determined by the parameters evaluated: spike count, burst count, burst spike count, percentage of spikes in bursts and interbursts interval (Figure 5B–F; Appendix A). In general, it seems that NCALD reduction increased the spike height in all MN lines (Figure 5A). The control WTC11 MNs and the SMA MNs exhibited a significant increase of spike numbers (Figure 5B) and burst numbers (Figure 5C) but not overall number of burst spike counts (Figure 5D) upon *NCALD*-ASO69 compared to CTRL-ASO treatment. Furthermore, the control HUVEC line and the SMA type I lines exhibited an increase in the percentage of spikes (Figure 5E) that are present in a burst after treatment with 60 nM *NCALD*-ASO69. Finally, *NCALD*-ASO69 had a strong impact in the reduction of the interval between neuronal bursts in the SMA type I line, HGK1 (Figure 5F). These results further support the relevant role of NCALD in neuronal activity and synaptic connectivity, and are consistent with our previous results showing amelioration of the electrophysiological properties of the gastrocnemius muscle in SMA mice upon *Ncald*-ASO re-injection treatment. Notably, the SMA type I line, HGK1 seems to be more sensitive to NCALD reduction than the SMA type I line, CS32iSMA, and a similar phenomenon is observed between the control MNs, HUVEC and WTC11. These data emphasize the importance of testing compounds in hiPSC-derived MNs, which better resemble the human genetic diversity, in order to provide a better understanding of which patients might respond to a treatment and which ones will most probably not.

## 3. Discussion

Since the discovery of the SMA disease-causing gene in 1995 [6], there have been significant advances in the understanding of the complex pathomechanisms underlying SMA. Most importantly, all the efforts facilitated the development of FDA- and EMA-approved therapies for SMA that changed the natural history of the disease [3,4,49]. Development of drugs that enhance SMN levels are, without any doubt, the most straightforward therapeutic strategy for SMA, and have shown impressive results in patients. However, even when the therapy is administered at pre-symptomatic stages, SMN-enhancing compounds might be still insufficient to completely counteract disease progression [23,24,25]. Moreover, a recent study has demonstrated that long-term overexpression of AAV9-SMN1 in a SMA mouse model induces a dose-dependent late-onset motor dysfunction characterized by loss of proprioceptive synaptic transmission and neurodegeneration. These observations highlight how crucial it is to understand the temporal requirements of SMN life-long: highest during neonatal life while NMJs undergo maturation and lower during adulthood [20,50]. Combinatorial approaches targeting SMN-dependent and -independent pathways disturbed in SMA might ameliorate functions or symptoms that cannot be improved by the increase of SMN solely. In this regard, SMA protective modifiers represent a unique opportunity to further ameliorate or rescue SMA independently of SMN upregulation. Indeed, pharmacological reduction of the genetic modifier NCALD using ASOs in combination with suboptimal dose of *SMN*-ASO has shown to ameliorate SMA pathology hallmarks in SMA mice at PND21. However, the therapeutic effect of *Ncald*-ASOs was rather short due to short-time stability of *Ncald*-ASOs [30].

In the present study, our main goal was to determine if *Ncald*-ASO re-injection could prolong the therapeutic effect observed at PND21 in SMA mice. Moreover, we wanted to assess the therapeutic effect of human *NCALD*-ASOs in hiPSC-derived MNs from control individuals and SMA type I patients. In summary, our main findings demonstrate: (i) Re-injection of 500 µg *Ncald*-ASO via i.c.v. bolus injection was well tolerated by the animals and significantly reduced NCALD levels in the CNS; (ii) Combinatorial therapy with *Ncald*-ASO re-injection in SMA mice ameliorates electrophysiological defects and denervation in the long-term compared to a single injection; (iii) Newly developed human *NCALD*-ASOs significantly downregulated NCALD in hiPSC-derived MNs, but only *NCALD*-ASO69 was well tolerated; (iv) *NCALD*-ASO69 has a positive impact on growth cone cytoskeleton dynamics of SMA type I MNs; (v) *NCALD*-ASO69 increased neuronal activity in SMA type I MNs.

### 3.1. Ncald-ASO Re-Injection Prolongs Amelioration of Electrophysiological Defects and NMJ Pathology in SMA Mice

Electrophysiological measurements (CMAP, MUNE) of the gastrocnemius muscle were significantly reduced in various SMA animal models even upon injection with low-dose *SMN-*ASO [30,37,51], implying a decreased motor functionality. In the previous combinatorial therapy approach, using *Ncald*-ASO and low-dose *SMN*-ASO, one single injection of *Ncald*-ASO at PND2 was not sufficient to ameliorate electrophysiological defects long-term [30]. Importantly, here, we demonstrated that NCALD reduction achieved by re-injection of *Ncald*-ASO at PND28, significantly increased CMAP amplitude and motor unit numbers in a long-term fashion. These results indicate that long-term NCALD reduction not only improves the functionality and number of motor units, but also prevents MNs from degeneration. Consequently, preserving neuromuscular function over time most probably halts the progression of the disease, alleviates muscle atrophy and improves muscle function [52]. Interestingly, SMN reduction leads to selective vulnerability of MN pools and muscles [53]. Therefore, it would be essential to investigate the effect of NCALD reduction by analyzing electrophysiological parameters of other affected muscles, for example, the proximal muscle quadratus lumborum.

NMJ denervation and loss of function are key hallmarks of SMA and other neuromuscular disorders such as ALS [54] or Myasthenia gravis [55] that lead to skeletal muscle atrophy. NMJs form during embryonic development and undergo complex maturation steps even following birth [52]. In SMA, MNs and NMJs show early pathological defects, including altered morphology and function that precede MN death. Importantly, MN loss is an irreversible pathogenic event, while NMJs have a strong plasticity and can further improve their functionality by axonal sprouting. In the present work, NMJ area from the TVA muscle was not increased after re-injection with *Ncald*-ASO compared to NMJ area of animals injected with CTRL-ASO. Instead, long-term treatment of SMA animals with *Ncald*-ASO showed a significant rescue of the NMJ denervation when compared to animals treated with CTRL-ASO. These results further suggest that the observed amelioration in the electrophysiological parameter CMAP is probably due to an increase of NMJs that are fully innervated and functional. Moreover, the MUNE results indicate that NCALD reduction seems to protect against denervation at the NMJ level (a dying-back phenomenon), which allows the maintenance of motor unit numbers over time. This evidence strongly emphasizes the therapeutic role of NCALD reduction in the NMJ pathology, and the importance of re-injections to prolong the beneficial effect.

### 3.2. NCALD-ASO69 Treatment Improves Cytoskeleton Dynamics and Neuronal Activity in hiPSC-Derived MNs

Contrary to *SMN*-ASO, which could be developed and then used in both, SMA mice which carry the human *SMN2* gene and in SMA patients [49,56,57], the orthologous *NCALD* genes differ between mice and humans and therefore, human-specific *NCALD*-ASO had to be developed and tested. We designed, developed and tested a full battery of *NCALD-*ASO and selected the three best hits showing the least toxicity and highest efficacy to reduce NCALD levels in various cell types. However, when testing these in MNs differentiated from hiPSC lines, only *NCALD*-ASO69 proved to be non-toxic and well tolerated. 

MNs develop very long axons with distal growth cones which are highly plastic and constantly changing structures dependent of cytoskeleton dynamics that during neuronal development respond to intra- and extra-cellular cues and are the driving force of axonal outgrowth towards the target [58,59]. Actin cytoskeleton dynamics plays an important role in neurodegeneration through involvement in axonal functions and synapse maintenance [60]. In SMA, SMN reduction leads to axonal outgrowth defects reduced growth cone size and defects in local translation of different actin isoforms, which results in defective actin dynamics [43,44]. Interestingly, NCALD binds to key components of the cytoskeleton, actin and tubulin [46], and its reduction promotes neurite outgrowth [29]. Since MN growth cones represent such a critical component of the future synapse with the skeletal muscle, and MN degeneration implies NMJ pathology [61], we analyzed growth cone morphology upon *NCALD*-ASO69 treatment in more detail. Analysis of actin morphology revealed a significant reduction of the number of blunt growth cones and an increase in filopodia and lamellipodial terminal ends upon *NCALD*-ASO69 treatment. These results demonstrate that NCALD reduction has a positive impact on growth cone actin dynamics and decreases the number of neurons with defective axonal outgrowth. Next, microtubule morphology was categorized according to the shape in the central domain of the growth cone: bundled (characterized by thin static microtubules), spread or looped [48]. Excitingly, *NCALD*-ASO69 treatment resulted in significant increase of spread and looped conformations in both control and SMA hiPSC-derived MNs compared to CTRL-ASO-treated MN cultures, which goes in line with the findings on actin morphology. In addition, this data strongly corroborates the protective role of NCALD reduction at the NMJ level observed in the mouse model.

Furthermore, we investigated if *NCALD*-ASO69 treatment has an impact on neuronal activity of MN cultures, since decreased neuronal firing rate is one SMA hallmark, which not only affects MNs but the overall neuronal circuits for motor control [62,63,64,65]. As expected, SMA MNs exhibited reduced spontaneous neuronal activity measured with the multielectrode array at day 20 of the differentiation when compared to control MNs. Moreover, pharmacological NCALD reduction had a significant increase in parameters associated with neuronal activity such as spike count, burst count and percentage of spikes in a burst in SMA and control MNs. In addition, the time between bursts was significantly reduced in *NCALD*-ASO69-treated SMA MNs. Interestingly, MNs of the control WTC11 and the SMA HGK1 lines responded better to the treatment than MNs from HUVEC and CS32iSMA. These data emphasize the importance of integrating hiPSC models in the preclinical studies, since hiPSC-derived from patients represent the full array of human genetic variability.

In conclusion, NCALD reduction in hiPSC-derived MNs increases neuronal activity, and supports the results obtained in the preclinical study, where the combination of *SMN*- and *Ncald*-ASOs ameliorate electrophysiological defects in SMA mice.

## 4. Materials and Methods

### 4.1. Mouse Model and Genotyping

The severely-affected Taiwanese SMA mouse model [FVB.Cg-Tg (*SMN2*)2Hung *Smn1*tm1Hung/J, stock number 005058] [66] was purchased from Jackson Laboratory (Bar Harbor, ME, USA). These homozygous SMA Taiwanese mice were originally on congenic FVB/N background, but we backcrossed them for >7 generations with C57BL6/N wild-type mice to obtain a congenic C57BL6/N background [67]. An intermediate mouse model was produced by crossing homozygous females on FVB/N background with male C57BL6/N *Smn*^ko/wt^ to generate the mixed50 SMA mice. This breeding results in a F1 offspring of ~50% severe SMA (*Smn*^ko/ko^; *SMN2*^tg/0^) mice and the corresponding healthy HET (*Smn*^ko/wt^; *SMN2*^tg/0^) mice were generated [12]. Here, we generated a mild SMA mouse model by subcutaneously injecting the F1 offspring with a suboptimal dose of 30 µg S*MN*-ASO in the skin fold of the neck at PND1 using a microliter syringe (Hamilton) [30,31,37]. Approximately equal numbers of male and female mice were used for all the experiments. For genotyping, the primers used were as follows: *Smn*^KO^fw: 5′-ATAACACCACCACTCTTACTC-3′; *Smn*^KO^rev1: 5′-AGCCTGAAGAACGAGATCAGC-3′; *Smn*^KO^rev2: 5′-TAGCCGTGATGCCATTGTCA-3′. Animals were maintained under a 12 h light/dark cycle with access to food and water ad libitum. Breeding, housing, and experimental use of animals were performed in a pathogen-free environment. All mouse experiments were approved by the local animal protection committee LANUV NRW (Landesamt für Natur, Umwelt und Verbraucherschutz) under the reference number 81-02.04.2019.A138.

### 4.2. Antisense Oligonucleotides (ASOs)

ASOs were designed and synthesized by IONIS Pharmaceuticals. Lyophilized stocks of *SMN*-ASO (ATTCACTTTCATAATGCTGG) were reconstituted with 1X PBS and stored in 10 mg/mL at −20 °C for in vivo injections. As a control for the i.c.v. treatment, we used CTRL-ASO (GTTTTCAAATACACCTTCAT). To downregulate *Ncald* in the CNS, the previously described 5-10-5 2′-O-methoxyethyl (MOE) gapmer with mixed PS/PO backbone *Ncald*-ASO was used (GTGGTTCTTGTTTTACAGGA) [30]. For in vitro treatment of hiPSC-derived MNs, the following 3-10-3 constrained ethyl (cEt)-modified gapmer *NCALD*-ASOs were used: *NCALD*-ASO55 (GACAGATATGACTTCC), *NCALD*-ASO69 (CACATAGATTAAACCA), *NCALD*-ASO89 (TCTTTTTGGTCTACCA) and CTRL-ASO (GGCCAATACGCCGTCA).

### 4.3. ASOs Injection In Vivo

Neonatal mice (PND2) were i.c.v. injected in the right hemisphere, between the confluence of sinuses with 100 µg of *Ncald*-ASO or CTRL-ASO using a glass needle [68]. ASOs concentration was determined photometrically (AD260) in order to administer 1.5 µL (1 µL ASO + 0.5 µL of 0.05% *w*/*v* trypan blue in PBS). For *Ncald*-ASO re-injections, i.c.v. bolus injection was performed at PND28. Animals were anesthetized using Ketamine (Ketaset 100 mg/mL, WDT Wirtschaftsgenossenschaft deutscher Tierärzte)/Xylazine (2%, Serumwerk Bernburg AG, Bernburg, Germany) and placed in a stereotaxic instrument (Bilaney, Cat# DKI940) with a mouse adaptor (Bilaney, Cat# DKI926) and a micropositioner (Bilaney, Cat#DK5000) using a thermostatic warming plate to maintain the body temperature during the procedure. Next, injection coordinates were determined from the mouse bregma (For HET: X = 1 mm, Y = 0.3 mm; for SMA X = 0.980 mm, Y = 0.250 mm; in all the Z = range between −1.6 mm and −1.7 mm) [69]. To drill a hole in the skull, we used a microdrill (burr size: 0.8 mm, KF Technology). A total of 5 µL of ASO was delivered at a rate of 1 µL/30 s.

### 4.4. Experimental Design

All offspring of each litter were blinded-injected and processed by randomized numbering. The experimenter was blinded regarding genotypes of the mice at all steps until final statistical analysis. Experimenter was blinded regarding cell line genotypes during image acquisition, recordings and statistical analysis. We conducted all experiments at least in triplicates. Multielectrode array experiments with hiPSCs correspond to three independent differentiations, each of them with at least three technical replicates. Blinding was performed by randomized numbering by an independent person. To avoid pseudo replication, the mean value per animal and mean value of technical replicates was applied for statistical analysis.

### 4.5. Western Blot

Brain and spinal cord were collected and lysed in RIPA buffer (Sigma, St. Louis, MO, USA) containing protease inhibitors (Complete Mini, Roche, Basel, Switzerland). The primary antibodies used: anti-beta-actin HRP-conjugated (Cat# HRP-60008, RRID:AB_2819183 Proteintech, Rosemont, IL, USA ), anti-ISL1 (mouse, Cat# 39.4D5, RRID:AB_2314683 Hybridoma Bank, Iowa City, IA, USA), anti-NCALD (rabbit, Cat# 12925-1-AP, RRID:AB_2149410 Proteintech, Rosemont, IL, USA), anti-SMN (mouse, Cat# 610646, RRID:AB_397973 BD Biosciences, San Jose, CA, USA). Signal was detected with rabbit-HRP-conjugated secondary antibody (Cat# 7074, RRID:AB_2099233 Cell Signaling Technology, Danvers, MA, USA) and mouse-HRP-conjugated secondary antibody (α-mouse, Cat# 115-035-003, RRID:AB_10015289 Jackson ImmunoResearch Labs, West Grove, PA, USA) and the chemiluminescence reagent (Thermo Fisher Scientific, Waltham, MA, USA).

### 4.6. Compound Muscle Action Potential and Motor Unit Number Estimation

Compound muscle action potential (CMAP) and motor unit number estimation (MUNE) were recorded as previously described [51]. First, mice were anesthetized with inhaled isofluorane (1 L/min O_2_ flow rate, 5% isofluorane for induction, 1.5–2% maintenance) and temperature was maintained at 37 °C with a thermostatic warming plate. Stimulation of the sciatic nerve at the proximal hind limb was achieved by placing the anode subcutaneously over the sacrum and the stimulation electrode (cathode) positioned subcutaneously at the sciatic notch. The active recording electrode (E1) was located subcutaneously at the proximal gastrocnemius muscle and the reference electrode (E2) was set at the metatarsal region of the right foot. Next, square-wave pulses of 0.1 ms with a <10 mA of intensity were applied to stimulate the sciatic nerve. Five repetitive CMAP responses were recorded per animal and the peak-to-peak amplitude was measured. To estimate MUNE, ten consecutive increments from the initial response were recorded and calculated as previously reported [30,51].

### 4.7. Analysis NMJ from the Transversus Abdominis (TVA)

The TVA muscle was fixed in 4% PFA for 20 min and stained according to standard immunohistochemistry protocols. Primary antibodies used: rabbit anti-NF-L (Cell Signaling Technology Cat# 2837, RRID:AB_823575), BTX-594 (Thermo Fisher Scientific Cat# B13423, RRID:AB_2617152). Secondary antibody rabbit AlexaFluor-488 (Thermo Fisher Scientific Cat# A-21206, RRID:AB_2535792). Muscles were mounted on microscope slides with Prolong Gold antifade reagent. NMJ size was quantified using ImageJ (RRID:SCR_003070). NMJ innervation was categorized according to the percentage of NF that innervates each end plate (BTX-positive). Equal or more than 80% was considered fully innervated, between 15% and 79% was counted as partially innervated and less than 15% denervated.

### 4.8. Muscle Fiber Analysis

Gastrocnemius muscles were embedded in paraffin and sectioned as described in [67]. Next, haematoxylin (#MHS32 Sigma-Aldrich, St. Louis, MO, USA) and eosin (#EYQ999 ScyTek Laboratories, Logan, MO, USA) staining was performed and muscle fiber area was determined with the ZEN software (RRID:SCR_013672) (Zeiss, Jena, Germany).

### 4.9. hiPSC Maintenance and Differentiation into MNs

Human iPSCs were routinely cultured in feeder-free conditions using Matrigel™-coated plates with mTeSR1 pluripotent media (StemCell Technologies, Vancouver, BC, Canada) supplemented with Pen/Strep. On the first day, mTeSR1 media was supplemented with 10 µM Y-27632 (S1049, Selleckchem, Houston, TX, USA) to enhance the survival of dissociated cells after passaging or cryopreservation. At 80–90% confluency, hiPSCs were either passaged to a new plate in lower density, cryopreserved or used to start a new MN differentiation. For hiPSC differentiation into MNs, we followed a protocol previously published [70] with some modifications. First, similar size squares (1–2 mm) were generated using a flame-modified Pasteur glass pipette in order to produce the clumps for embryoid body formation and cultures were treated with 3 mg/mL Collagenase type IV (Gibco, Thermo Fisher Scientific) at 37 °C. Clumps were gently resuspended in EB formation media containing Essential™6 (Gibco, Thermo Fisher Scientific) medium supplemented with 10 µM Rock inhibitor and transferred to ultra-low attachment flasks (Corning, Corning, NY, USA) on an horizontal shaker (40 rpm) in the incubator at 37 °C. For the first two days of the differentiation, neuronal basal media (DMEM/F12 Glutamax and Neurobasal supplemented with N2 and B27 without vitamin A) was supplemented with 3 µM CHIR99021 (4423, Tocris Bioscience, Bristol, UK), 0.2 µM LDN-193189 (S2618, Selleckchem), 40 µg SB431542 (1614, Tocris Bioscience), and 5 µM Y-27632 (S1049, Selleckchem). From day 3 on, neuronal basal media was supplemented with 0.1 µM retinoic acid (Sigma) and 500 nM SAG (Merck Millipore, Burlington, VT, USA). From day 8 on until the end of the differentiation BDNF (10 ng/mL, Peprotech, Waltham, MA, USA) and GDNF (10 ng/mL, Peprotech) were added to the media. From day 9 to 11, neuronal media was supplemented with 10 µM DAPT (2634, Tocris Bioscience), and from day 12–16 with 20 µM DAPT. Until day 11 included, media was changed every day. On day 11, clumps with MN progenitors were dissociated into single cells for plating on 20 µg/mL laminin coated wells. Every day, half of the media was changed. From day 17 on, maturation media containing exclusively 10 ng/mL of BDNF, GDNF and CNTF (Peprotech) was added. Every other day, media was changed by replacing half of the medium.

### 4.10. NCALD-ASOs Treatment in hiPSC-Derived MNs

Cells were plated in a density of 1.2 × 10^6^ cells in 6-well and 1 × 10^5^ for multielectrode array experiments. Next, cells were transfected at day 13 with different concentration of *NCALD*-ASOs or CTRL-ASO using Lipofectamine 3000 transfection reagent, according to the manufacturer’s protocol. 

### 4.11. hiPSCs and MNs Immunohistochemistry

Immunofluorescent stainings of cells were conducted using a standard protocol. For fixation, 4% PFA and 4% PFA supplemented with 4% sucrose was applied to hiPSCs and hiPSC-derived MNs, respectively, for 10–15 min at room temperature. The primary antibodies used Anti-OCT 3/4 (mouse, Cat# sc-5279, RRID:AB_628051 Santa Cruz Biotechnology, Dallas, TX, USA), anti-SOX2 (mouse, Santa Cruz Cat# sc-365823, RRID:AB_10842165), anti-SMN (mouse, BD Biosciences Cat# 610646, RRID:AB_397973), anti-ISL1 (mouse, Hybridoma Bank Cat# 39.4D5, RRID:AB_2314683), anti-TUJI (rabbit, Cat# ab18207, RRID:AB_444319 Abcam, Cambridge, UK) and Phalloidin conjugated AlexaFluor568 (Thermo Fisher Scientific; A12380). Secondary antibodies used anti-rabbit AlexaFluor488 (Thermo Fisher Scientific Cat# A-21206, RRID:AB_2535792), anti-mouse AlexaFluor568 (Thermo Fisher Scientific Cat# A10042, RRID:AB_2534017).

### 4.12. Multielectrode Array

Electrophysiological recordings at day 20 of the differentiation were performed with a multi-electrode array (Multichannels System, Reutlingen, Germany) using a 24 well plate containing 12 gold electrodes per well, as recently described [71]. Briefly, plates were coated using Poly-L-ornithine and 20 µg/mL laminin. At day 11 of the differentiation, after EBs dissociation, a total of 100,000 cells were seeded in each well, diluted in 60 µL of MN media day 11. Cells were left to settle for 2 h before adding 1 mL of the corresponding MN media, in an incubator at 37 °C and 5% CO. On day 13 of the differentiation, MNs were treated with 60 nM CTRL-ASO or *NCALD*-ASO69 using Lipofectamine 3000. Importantly, measurements at day 20 of the differentiation were taken before media change, since addition of fresh media transiently alters the electrophysiological properties of the MNs. Before each recording, the plate was left for 2 min on the recorder. Experimental recordings took place in a non-humidifier incubator at 37 °C for 3 min. Data acquisition and analysis was performed with the Multi-Channel Suite software (Multichannels system) applying the following parameters: threshold for spike detection ± 10 µV, band pass filter with 100 Hz and 3 kHz cut-off frequencies. In order to detect the spikes, an adaptive threshold at 5.5 times the standard deviation of the estimated noise on each electrode was set. Three independent differentiations per line were considered for the statistical analysis.

### 4.13. Image Acquisition and Analysis

Fluorescence images of NMJs and cells were acquired with Zeiss microscope (AxioImager.M2) supplied with the Apotome.2 system to mimic a confocal microscope. Images were acquired as Z-stacks with 20×, 40× and 63× objectives. The quantitative analysis was performed with ZEN (RRID:SCR_013672) (Zeiss) and Fiji software. Bright field images for muscle fiber analysis were acquired with Zeiss microscope (Axioskop.2) equipped with AxioCamICc1 and 20× objective. 

### 4.14. Image Acquisition and Analysis

Fluorescence images of NMJs and proprioceptive inputs were acquired with Zeiss microscope (AxioImager.M2) supplied with the Apotome.2 system to mimic a confocal microscope. Images were acquired as Z-stacks with 20×, 40× objectives. The quantitative analysis was performed with ZEN (RRID:SCR_013672) (Zeiss) and Fiji software. Bright field images for muscle fiber analysis were acquired with a Zeiss microscope (Axioskop.2) equipped with AxioCamICc1 and 20× objective. Brain sections were imaged with a Leica Slide Scanner (SCN400).

### 4.15. Statistics

Statistical analysis was performed using the software programs MS Excel 2016 (Microsoft, Redmond, WA, USA) and GraphPad Prism 9 (GraphPad Software, San Diego, CA, USA). Unpaired, two tailed Student’s t-tests and one-way ANOVA statistics tests with Tukey post hoc test for multiple comparisons were performed. To compare categories (NMJ innervation and growth cone analysis) χ^2^ test was performed, and for more than two comparisons, for normalization, the p value was multiplied by the total number of comparisons. The figure legends depict the specific statistical test used, sample size and *p*-values, respectively.

## 5. Conclusions

Our data here further support the protective effect of NCALD reduction on MN and NMJ function in SMA in mice and in human cellular system. Among a full battery of newly developed human *NCALD*-ASOs, we identified one that is non-toxic and efficiently downregulates NCALD in human MNs. It restores the cytoskeletal dynamics and neuronal activity in MN differentiated from SMA hiPSCs to control level. This opens the avenue for combinatorial therapies in SMA patients.

## 6. Patents

BW holds a US patent 9,988,626 B2 approved 5 June 2018, Neurocalcin Delta Inhibitors and Therapeutic and Non-Therapeutic Uses Thereof.

## Figures and Tables

**Figure 1 ijms-24-04198-f001:**
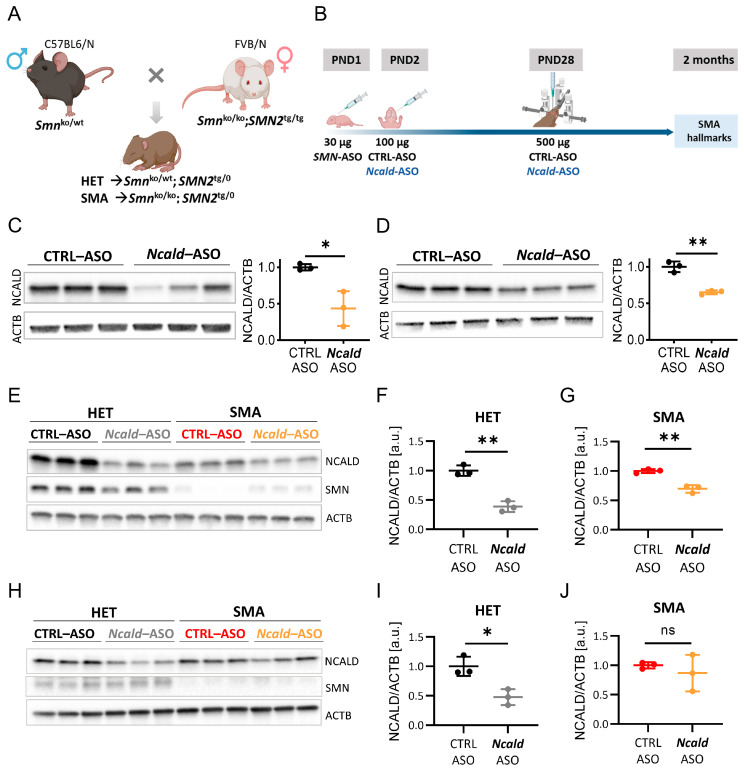
Experimental scheme and efficacy of 500 µg *Ncald*-ASO i.c.v bolus re-injection in adult mice. (**A**) Breeding strategy to obtain intermediate SMA and HET mice on mixed50 background. Figure created with BioRender.com. (**B**) Scheme showing combinatorial therapy of SMA and HET mice (latter used as controls) with *SMN-* and either CTRL- or *Ncald*-ASOs. Figure created with BioRender.com. (**C**,**D**) Western blot analysis and quantification of NCALD levels in brain (**C**) and spinal cord (**D**) lysates from CTRL- or *Ncald*-ASO-treated wild-type mice at PND28; samples were collected 2-weeks after injection. Two-weeks post i.c.v bolus injection with 500 µg *Ncald*-ASO, NCALD levels in the brain and spinal cord are significantly reduced compared to CTRL-ASO-treated animals. Unpaired two-tailed Student‘s *t*-test. All values reported as mean and error bars represent ± SD. * *p* ≤ 0.05, ** *p* ≤ 0.01. Each dot in the bar graphs represents an independent animal. (**E**–**G**) Western blot analysis and quantification of NCALD levels in brain lysates from ASO-treated experimental animals. Samples were collected 4-weeks after treatment. Four-weeks post i.c.v bolus injection with 500 µg *Ncald*-ASO, HET and SMA mice showed significantly reduction of NCALD amount compared to CTRL-ASO-treated animals. Unpaired two-tailed Student‘s *t*-test. All values reported as mean and error bars represent ± SD. ** *p* ≤ 0.01. Each dot in the bar graphs represents an independent animal. (**H**–**J**) Western blot analysis and quantification of NCALD levels in spinal cord lysates from ASO-treated experimental animals. Samples were collected 4-weeks after i.c.v bolus injection at PND28 with 500 µg *Ncald*-ASO, HET but not SMA mice showed significantly reduced NCALD protein amount compared to CTRL-ASO-treated animals. Unpaired two-tailed Student‘s *t*-test. All values reported as mean and error bars represent ± SD. * *p* ≤ 0.05. Each dot in the bar graphs represents an independent animal. ns = not significant.

**Figure 2 ijms-24-04198-f002:**
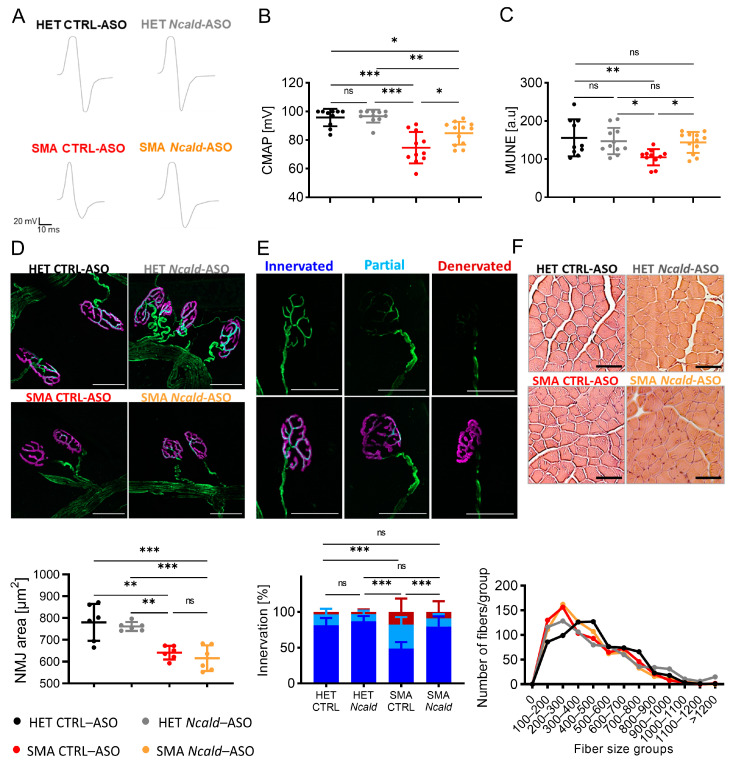
Re-injection with 500 µg *Ncald*–ASO improves electrophysiological defects and NMJ innervation at 2 months. (**A**) Representative traces of CMAP response in HET and SMA mice in the gastrocnemius muscle at 2 months of age, 4-weeks after re-injection. (**B**) Quantification of sciatic CMAP response of the gastrocnemius muscle in the four analyzed experimental groups. For CMAP response, peak-to-peak amplitude was quantified. Each dot in the graph represents one animal, N = 10–12. Ordinary one-way ANOVA with Tukey post hoc test for multiple comparisons. Error bars represent ± SD. * *p* ≤ 0.05, ** *p* ≤ 0.01, *** *p* ≤ 0.001. (**C**) Quantification of the gastrocnemius muscle MUNE in the four analyzed experimental groups. Each dot in the graph represents one animal, N = 10–12. Ordinary one-way ANOVA with Tukey post hoc test for multiple comparisons. Error bars represent ± SD. * *p* ≤ 0.05, ** *p* ≤ 0.01. (**D**) Transversus abdominis NMJ area. Representative images and quantification of NMJ area (below) at 2 months of age, showing postsynaptic NMJ region (BTX, magenta) and presynaptic nerve (NF, green) (scale bar: 50 μm). NMJ area was analyzed with ImageJ (N = 6, n = 100 NMJs/mouse). Statistics were performed with mean values of animals per group. Ordinary one-way ANOVA with Tukey post hoc test for multiple comparisons, ** *p* ≤ 0.01, *** *p* ≤ 0.001. (**E**) Transversus abdominis NMJ innervation. Representative images of TVA NMJ innervation in each category, showing the postsynaptic region (BTX, magenta) and presynaptic nerve (NF, green). NMJ innervation was classified in three categories according to the degree of innervation. Quantification (below) of all four experimental groups. HET CTRL-ASO (N = 5, n = 272), HET *Ncald*-ASO (N = 5, n = 270), SMA CTRL-ASO (N = 5, n = 259), SMA *Ncald*-ASO (N = 6, n = 322). Results are presented in percentages. *** denotes statistical significance *p* ≤ 0.001 (χ^2^ test). (**F**) Representative images of H&E staining and quantification (below) of gastrocnemius muscle fibers from HET and SMA animals at 2 months of age. Scale bar 50 µm. Gastrocnemius muscle size categorization according to area intervals of 100 µm^2^. In total, 5 animals were analyzed per genotype, and 100 muscle fibers per animal. ns = not significant.

**Figure 3 ijms-24-04198-f003:**
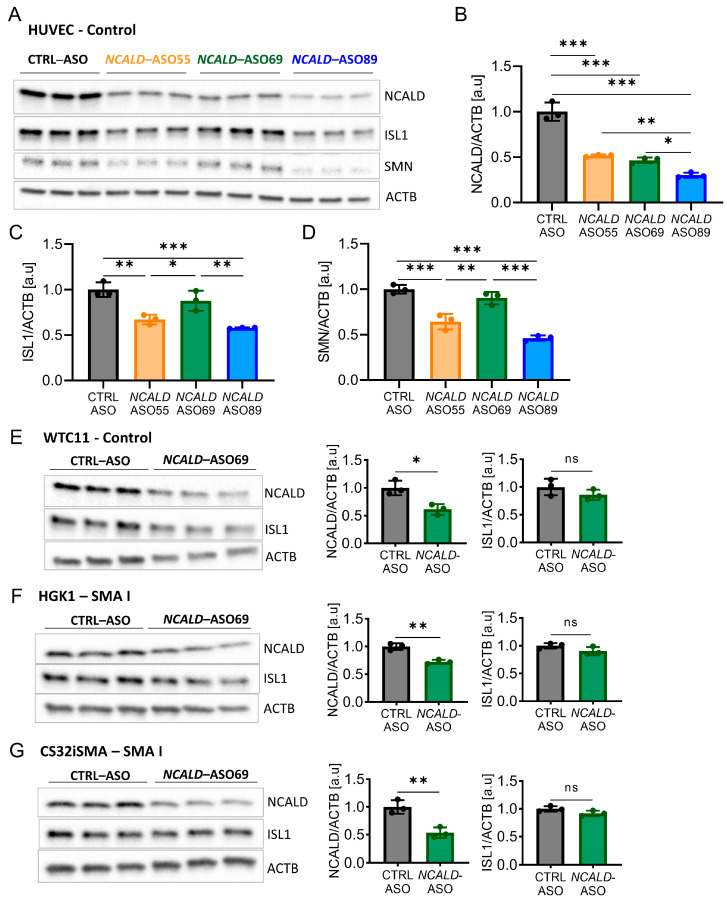
Efficiency and tolerability of *NCALD*-ASO candidates 7 days after treatment in hiPSC-derived MNs. (**A**) Western blot of HUVEC hiPSC-derived MNs treated with 60 nM of CTRL-ASO or the respective *NCALD*-ASO candidate using Lipofectamine™ at day 13, proteins collected at day 20 of the differentiation. ACTB used as loading control. (**B**) Quantification of NCALD levels in HUVEC MNs at day 20 after treatment with different *NCALD*-ASOs. Ordinary one-way ANOVA with Tukey post hoc test for multiple comparisons. (**C**) Quantification of ISL1 levels in HUVEC MNs at day 20 after treatment with each *NCALD*-ASOs. Ordinary one-way ANOVA with Tukey post hoc test for multiple comparisons. (**D**) Quantification of SMN levels in HUVEC MNs at day 20 after treatment with each *NCALD*-ASOs. Ordinary one-way ANOVA with Tukey post hoc test for multiple comparisons. (**E**–**G**) Western blots of control WTC11 (**E**), SMA type I HGK1 (**F**) and SMA type I CS32iSMA (**G**) MNs treated at day 13 of the differentiation with 60 nM of CTRL-ASO or *NCALD*-ASO69 using Lipofectamine™. Proteins were collected at day 20 of the differentiation. NCALD protein levels are significantly reduced upon *NCALD*-ASO69 treatment in all lines. No significant reduction of ISL1 protein and thus impact on MNs was detected. Unpaired two-tailed Student’s *t*-test was performed. All values reported as mean and error bars represent ± SD. N = 3, each dot in the bar graphs represents an independent well. * *p* ≤ 0.05, ** *p* ≤ 0.01, *** *p* ≤ 0.001. ns = not significant.

**Figure 4 ijms-24-04198-f004:**
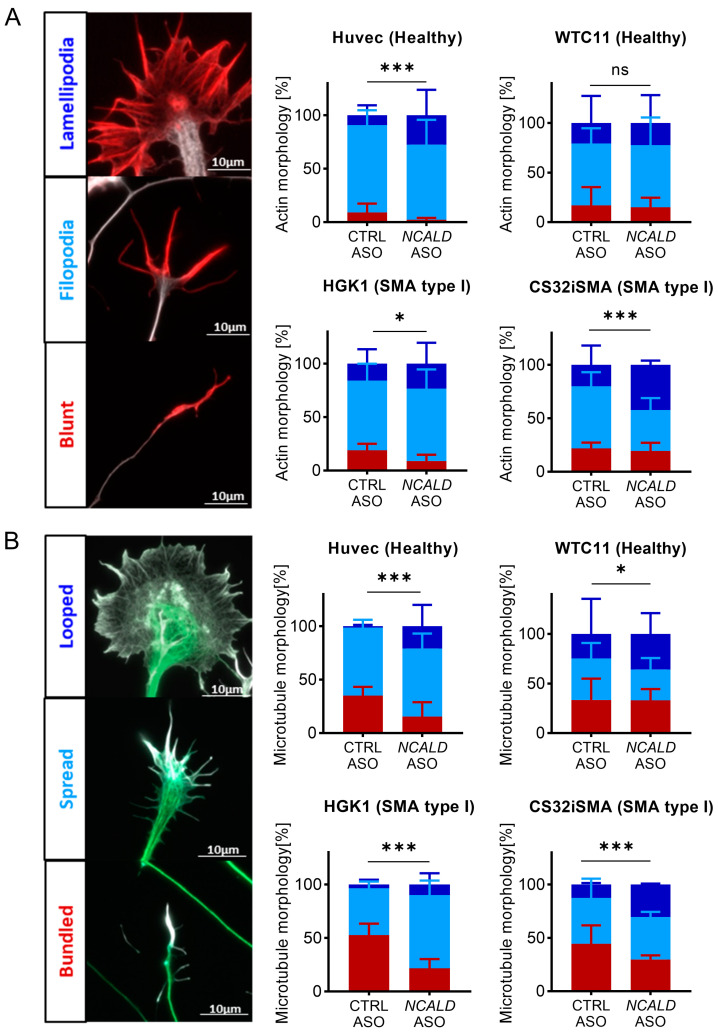
Analysis of cytoskeleton morphology upon *NCALD*-ASO69 treatment in hiPSC-derived MNs shows more mature growth cones. (**A**) For actin morphology analysis, actin was stained using phalloidin (red) and microtubules were stained using β-III-tubulin (TUJ1, white). Actin morphology was categorized as blunt ended, filopodia or lamellipodia based on the morphology of the filaments. Scale bar 10 µm. (**B**) Microtubules were stained using β-III-tubulin (TUJ1, green), and actin was counterstained using phalloidin (white). Microtubule morphology was categorized as bundled, spread or looped. Scale bar 10 µm. In total, three independent experiments from three independent differentiation were performed per cell line. HUVEC (N = 3; CTRL-ASO n = 141, *NCALD*-ASO69 n = 103), WTC11 (N = 3; CTRL-ASO n = 84, *NCALD*-ASO n = 85), HGK1 (N = 3; CTRL-ASO n = 138, *NCALD*-ASO69 n = 149), CS32isma (N = 3; CTRL-ASO n = 154, *NCALD-*ASO69 n = 146). Error bars show ± SD. Graphs represent percentages. * *p* ≤ 0.05, *** *p* ≤ 0.001 (Chi-square test). ns = not significant.

**Figure 5 ijms-24-04198-f005:**
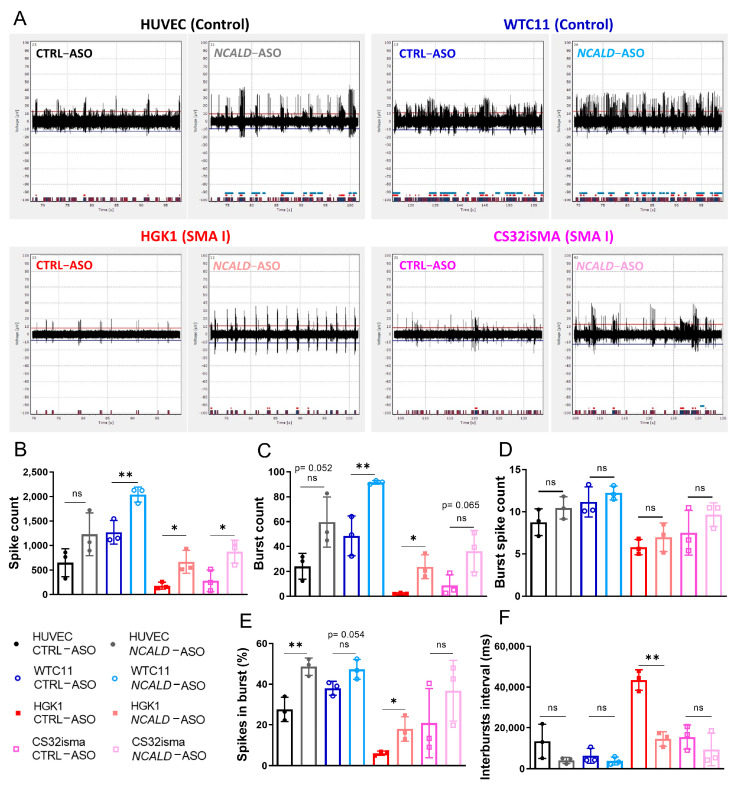
Multielectrode array shows increase in neuronal firing upon *NCALD*−ASO69 treatment. (**A**) Representative images of the activity detected by one electrode for 30 s per condition. (**B**–**F**) MEA analysis of various parameters (**B**) spike count, (**C**) burst count, (**D**) burst spike count, (**E**) spikes in burst and (**F**) interbursts interval measured at day 20 of CTRL-ASO or *NCALD*-ASO69-treated MNs. Dots in the graphs represent the average values of the technical replicates of each independent experiment, in total, three independent experiments corresponding to three independent MN differentiations (N = 3, n = 3–4). Unpaired two-tailed Student’s *t* test was performed in each line to compare CTRL-ASO versus *NCALD*-ASO treatment. All values reported as mean and error bars represent ± SD. * *p* ≤ 0.05, ** *p* ≤ 0.01. ns = not significant.

**Table 1 ijms-24-04198-t001:** Relevant information about hiPSC lines.

Cell Line	Phenotype	SMN1/SMN2 Copies	Sex	Age Sampling	Reprogrammed
HUVEC	Healthy control	2/2	male	fetal	Retrovirus
WTC11	Healthy control	2/2	male	30 years	Episomal plasmid
HGK1	SMA I	0/2	female	6 months	Retrovirus
CS32iSMA	SMA I	0/2	male	7 months	Episomal plasmid

hiPSC lines: Healthy control HUVEC hiPSC line was a generous gift from the Kurian Lab [41]. Healthy control WTC11 hiPSC line was generated by the Conklin Lab and purchased from Erasmus MC iPS Core Facility. SMA type 1 HGK1 hiPSC line was generated by iPIERIAN [42]. SMA type 1 CS32iSMA hiPSC line was purchased from Cedars Sinai iPSC Core Facility.

## Data Availability

No bioinformatic datasets were generated.

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
