# Peer review of "Long-Term SMN- and Ncald-ASO Combinatorial Therapy in SMA Mice and NCALD-ASO Treatment in hiPSC-Derived Motor Neurons Show Protective Effects"

_ijms, 2023, doi:10.3390/ijms24044198_

Round 1
Reviewer 1 Report
This study on SMA, which is a disease that requires serious developments in terms of treatment worldwide, is a very important study. Although the manuscript is suitable for publication even in its current state, I would like to make a few suggestions to the authors. By taking these suggestions into account, the authors can enhance the quality and importance of the manuscript.
- Keywords must be written in alphabetical order.
- The authors could add information about the financial loss related to SMA disease in the introduction section, which would highlight the importance of the manuscript better.
- The conclusion section doesn't emphasize the importance of the study enough. It would be appropriate to rewrite it to highlight the significance of the study.
Author Response
Thank you very much for this excellent feedback and positive review. It was a great pleasure to read it. As suggested by the reviewer:
- We have sorted the keywords alphabetically
- We have added a sentence to the financial burden in SMA and an additional reference
- We better highlighted the importance of this work in the conclusion
Reviewer 2 Report
Given the partial efficiency of SMN-dependent therapies in improving the symptoms caused by SMA in humans, as well as a short-lived, albeit improved response following a combination of SMN-ASOs with NCALD-ASOs in mice, the latter compound being administered at PND2, in this article the authors have aimed, and successfully managed, to demonstrate the improvement of the electrophysiological defects and NMJ innervation following the administration of an additional i.c.v. bolus in SMN and NCALD-ASO treated mice. Additionally, they have developed an efficient and safe for humans NCALD-ASO which improved the growth cone maturation and neuronal activity in SMA MNs.
The article is impeccably written and very well-documented, with enough scientific background in the introduction supported by strong and recent references. The design of the research is very carefully-thought and detail-oriented. The images have thorough and clear explanations. Overall, besides some minor typos in the first part of the article, this is a very valuable, original and high-quality work which can for sure be published without the need for any major improvement.
Author Response
Thank you very much for this excellent feedback and positive review. It was a great pleasure to read it.